# STREAMER: Streaming Representation Learning and Event Segmentation in a Hierarchical Manner

**Ramy Mounir**      **Sujal Vijayaraghavan**      **Sudeep Sarkar**

Department of Computer Science and Engineering, University of South Florida, Tampa

{`ramy, sujal, sarkar`}@usf.edu

## Abstract

We present a novel self-supervised approach for hierarchical representation learning and segmentation of perceptual inputs in a streaming fashion. Our research addresses how to semantically group streaming inputs into chunks at various levels of a hierarchy while simultaneously learning, for each chunk, robust global representations throughout the domain. To achieve this, we propose STREAMER, an architecture that is trained layer-by-layer, adapting to the complexity of the input domain. In our approach, each layer is trained with two primary objectives: making accurate predictions into the future and providing necessary information to other levels for achieving the same objective. The event hierarchy is constructed by detecting prediction error peaks at different levels, where a detected boundary triggers a bottom-up information flow. At an event boundary, the encoded representation of inputs at one layer becomes the input to a higher-level layer. Additionally, we design a communication module that facilitates top-down and bottom-up exchange of information during the prediction process. Notably, our model is fully self-supervised and trained in a streaming manner, enabling a single pass on the training data. This means that the model encounters each input only once and does not store the data. We evaluate the performance of our model on the egocentric EPIC-KITCHENS dataset, specifically focusing on temporal event segmentation. Furthermore, we conduct event retrieval experiments using the learned representations to demonstrate the high quality of our video event representations. Illustration videos and code are available on our project page: https://ramymounir.com/publications/streamer.

## 1 Computational theory

In temporal event analysis, an event is defined as "a segment in time that is perceived by an observer to have a *beginning* and an *end*" [59]. Events could be described by a sequence of constituent events of relatively finer detail, thus forming a hierarchical structure. The end of an event and the beginning of the next is a *segmentation boundary*, marking an event transition. Segmentation boundaries in the lower levels of the hierarchy represent event transitions at relatively granular scales, whereas boundaries in higher levels denote higher-level event transitions.

We propose a structurally self-evolving model to learn the hierarchical representation of such events in a self-supervised streaming fashion through predictive learning. Structural evolution refers to the model's capability to create learnable layers *ad hoc* during training. One may argue that existing deep learning architectures are compositional in nature, where high-level features are composed of lower-level features, forming a hierarchy of features. However, it is important to distinguish between a feature hierarchy and an event hierarchy: an event hierarchy is similar to a part/whole hierarchy in the sense that each event has clear boundaries that reflect the beginning and the end of a coherent chunk of information. One may also view the hierarchy as a redundancy pooling mechanism, where

37th Conference on Neural Information Processing Systems (NeurIPS 2023).

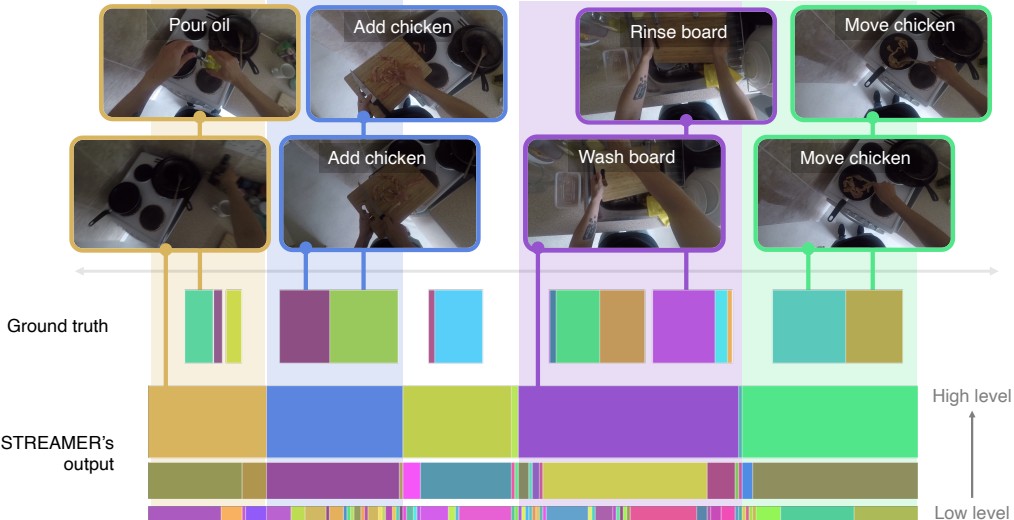

Figure 1: Comparison of STREAMER's hierarchical output to single-level ground truth annotations from EPIC-KITCHENS. The ground truth contains redundant narrations for successive annotations (*e.g.*, *add chicken* ■, ■); STREAMER identifies such instances as a single high level event (■). (Narrations from ground truth)

information grouped as one event is considered redundant for a higher level and can be summarized into a single representation for higher-level processing.

Our model is capable of generating a hierarchy of event segments (Figure 1) by learning unique semantic representations for each event type directly from video frames. This is achieved through predictive learning, which models the causal structure of events. These learned representations are expressive enough to enable video snippet retrieval across videos. Each level in the hierarchy selectively groups inputs from the level below to form coherent event representations, which are then sent to the level above. As a result, the hierarchy exhibits temporally aligned boundaries, with each level containing a subset of the boundaries detected in the lower level.

As often prescribed [28, 24], we impose the following biologically-plausible constraints on our learning algorithm:

1. The learning algorithm should be **continuous and online**. Most existing learning algorithms offer batch-based offline learning. However, the learning in the neocortex occurs continuously in a streaming fashion while seeing each datapoint only once

2. The learning should involve the ability to make **high-order predictions** by "incorporating contextual information from the past. The network needs to dynamically determine how much temporal context is needed to make the best predictions" [24] (Section 1.1)

3. Learning algorithms should be **self-supervised** and should not assume labels for training [37]; instead, they should be able to figure out the learning objective from patterns and causal structures within the data

4. The learning should stem from a **universal general-purpose algorithm**. This is supported by observations of the brain circuitry showing that all neocortical regions are doing the same task in a repeated structure of cells [24]. Therefore, there should be no need for a global loss function (*i.e.*, end-to-end training with high-level labels); local learning rules should suffice (Section 2)

## 1.1 Predictive learning

Predictive learning refers to the brain's ability to generate predictions about future events based on past experiences. It is a fundamental process in human cognition that guides perception, action, and thought [58, 31]. The discrepancy between the brain's predictions and the observed perceptual

inputs forms a useful training signal for optimizing cortical functions: if a model can predict into the future, it implies that it has learned the underlying causal structure of the surrounding environment. Theories of cognition hypothesize that the brain only extracts and selects features from the previous context that help in minimizing future prediction errors, thus making the sensory cortex optimized for prediction of future input [51]. A measure of intelligence can be formulated as the ability of a model to generate accurate, long-range future prediction [53].To this end, we design an architecture with the main goal of minimizing the prediction error, also referred to as maximizing the *model evidence* in Bayesian inference according to the free energy principle [19, 18].

Event segmentation theory (EST) suggests that desirable properties such as event segmentation emerge as a byproduct of minimizing the prediction loss [59]. Humans are capable of *chunking* streaming perceptual inputs into events (and *chunking* spatial regions into objects [14]) to allow for memory consolidation and event retrieval for better future predictions. EST breaks down streaming sensory input into chunks by detecting event boundaries as transient peaks in the prediction error. The detected boundaries trigger a process of transitioning (*i.e.*, shifting) to a new event model whereby the current event model is saved in the event schemata, and a different event model is retrieved, or a new one initialized to better explain the new observations. One challenge in implementing a computational model of EST is encoding long-range dependencies from the previous context to allow for contextualized representations and accurate predictions. To address this challenge, we construct a hierarchy of event models operating at different time-scales, predicting future events with varying granularity. This hierarchical structure enables the prediction function at any layer to extract context information dynamically from any other layer, enhancing prediction during inference (learning constraint 2). Recent approaches [42, 40, 41, 55] inspired by EST have focused on event boundary detection using predictive learning. However, these methods typically train a single level and do not support higher-order predictions.

## 1.2 Hierarchical event models

A single-level predictive model considers events that occur only at a single level of granularity rendering them unable to encode long-range, higher-order causal relationships in complex events. Conversely, a high-level representation does not contain the level of detail needed for accurately predicting low-level actions; it only encodes a high-level conceptual understanding of the action. Therefore, a hierarchy of event models is necessary to make predictions accurately at different levels of granularity [37, 26]. It is necessary to continuously predict future events at different levels of granularity, where low-level event models encode highly detailed information to perform short-term prediction and high-level event models encode conceptual low-detail features to perform long-term prediction.

EST identifies event boundaries based on transient peaks in the prediction error. To learn a hierarchical structure, we extend EST: we use event models at the boundaries in a layer as inputs to the layer above. The prediction error of each layer determines event demarcation, regulating the number of inputs pooled and sent to the layer above. This enables dynamic access to long-range context for short-term prediction, as required. This setup results in stacked predictive layers that perform the same prediction process with varying timescales subjective to their internal representations.

## 1.3 Cross-layer communication

As noted in Section 1.2, coarsely detailed long-range contexts come from higher layers (the top-left block of Figure 2), and highly detailed short-range contexts come from lower layers (the bottom-left block of Figure 2), both of which are crucial to predict future events accurately. Therefore, the prediction at each layer should be conditioned upon its own representation and those of the other layers (Equation (2)). These two types of contexts can be derived by minimizing the prediction error at different layers. Hence, making perfect predictions is not the primary goal of this model but rather continuously improving its overall predictive capability.

### 1.3.1 Contextualized inference

A major challenge in current architectures is modeling long-range temporal dependencies between inputs. Most research has focused on modifying recurrent networks [27, 10] or extending the sequence length of transformers [11, 57] to mitigate the problem of vanishing and exploding gradients

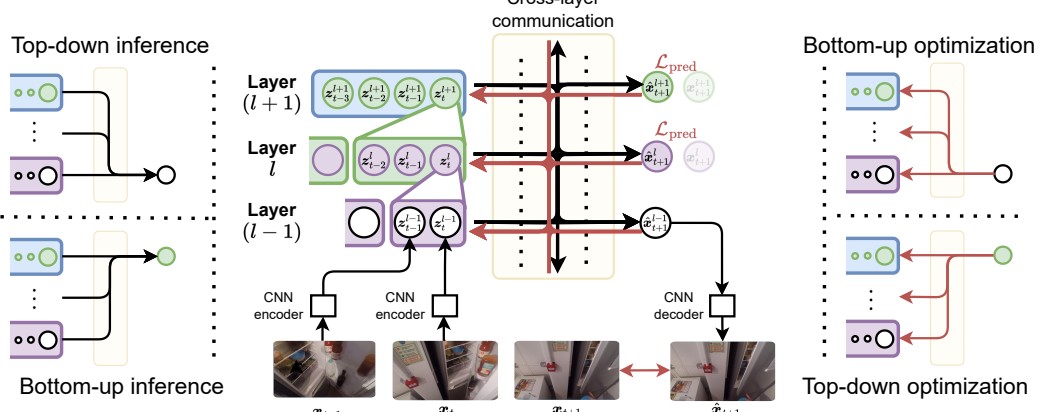

Figure 2: Given a stream of inputs at any layer, our model combines them and generates a bottleneck representation, which becomes the input to the level above it. The cross-layer communication could be broken down into top-down and bottom-up contextualized inference (left) and optimization (right).

in long-range backpropagation through time [44]. Instead, we solve this problem by allowing the multi-level context representations to be shared across layers during inference. It is worth noting that this type of inference is rarely used in typical deep learning paradigms, where the top-down influence only comes from backpropagating a supervised loss signal (*i.e.*, top-down optimization). Biologically-inspired architectures such as PredNet [39] utilize top-down inference connections to improve low-level predictions (*i.e.*, frames); however, these predictive coding architectures send the prediction error signal from each low-level observation (*i.e.*, each frame) to higher levels which prevents the network from explicitly building hierarchical levels with varying degrees of context granularity.

### 1.3.2 Contextualized optimization

Contextualized inference improves prediction, which is crucial for event boundary detection. However, we also aim to learn rich, meaningful representations. In Section 1, we noted that a 'parent' event could consist of multiple interchangeable low-level events. For instance, making a sandwich can involve spreading butter or adding cheese. From a high-level, using either ingredient amounts to the same parent event: "making a sandwich". Despite their visual differences, the prediction network must embed meaning and learn semantic similarities between these low-level events (*i.e.*, spreading butter and adding cheese).

We implement this through "contextualized optimization" of events (Section 2.2), where each layer aligns the input representations from the lower level to minimize its own prediction loss using its context. It must be noted that the contextualization from higher layers (Figure 2, bottom-right) is balanced by the predictive inference at the lower levels (Figure 2, top-right), which visually distinguishes the interchangeable events. This balance of optimization embeds meaningful representations into the distinct low-level representations without collapsing the model. These representations can also be utilized for event retrieval at different hierarchical levels (Figure 5). Unlike other representation learning frameworks that employ techniques like exponential moving average (EMA) or asymmetric branches to prevent model collapse [7, 21, 9], we ensure that higher layers remain grounded in predicting lower-level inputs through bottom-up optimization. [1]

## 2   Algorithm

Our goal is to incrementally build a stack of identical layers over the course of the learning, where each layer communicates with the layers above and below it. The layers are created as needed and are trained to function at different timescales; the output events from layer $l$ become the inputs to the layer $(l+1)$, as illustrated in Figure 3. We describe the model and its connections for a single layer $l$,

---

[1]A more detailed literature survey is in the Appendix

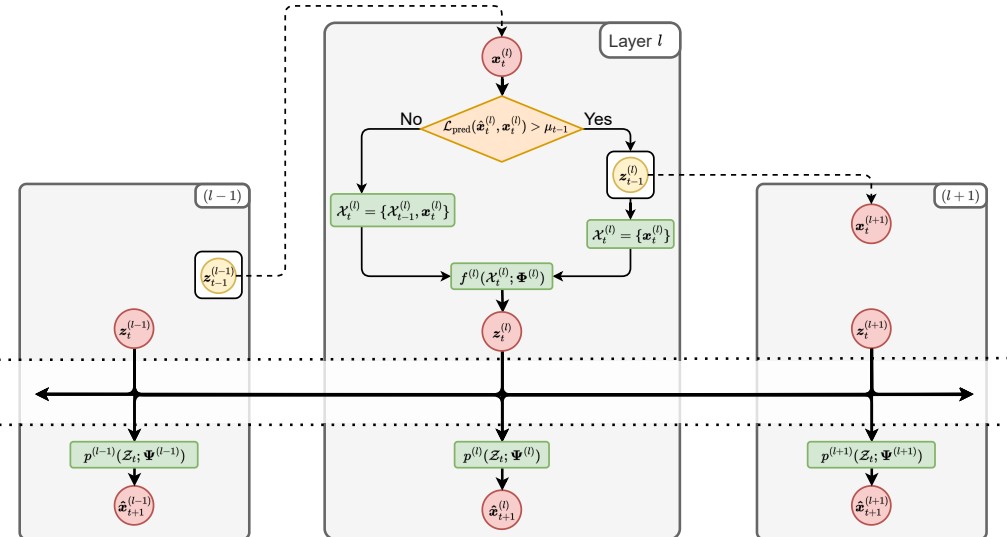

Figure 3: A diagram illustrating information flow across stacked identical layers. Each layer compares its prediction $\hat{\boldsymbol{x}}_t$ with the input $\boldsymbol{x}_t$ received from the layer below. If the prediction error $\mathcal{L}_{\text{pred}}$ is over a threshold $\mu_{t-1}$, the current representation $\boldsymbol{z}_{t-1}$ becomes the input to the layer above, and the working set is reset with $\boldsymbol{x}_t$; otherwise, $\boldsymbol{x}_t$ is appended to the working set $\mathcal{X}_t$

but the same structure applies to all the layers in the model (learning constraint 4). In what follows, we describe the design of a mathematical model for a single predictive layer that is capable of (1) encoding temporal input into unique semantic representations (the *event model*) contextualized by previous events, (2) predicting the location of event boundaries (event demarcation), and (3) allowing for communication with other existing layers in the prediction stack to minimize its own prediction loss.

## 2.1 Temporal encoding

Let $\mathcal{X}^{(l)} = \{\boldsymbol{x}^{(l)}_{(t-m)}, \ldots, \boldsymbol{x}^{(l)}_t\}$ be a set of $m$ inputs to a layer $l$ at discrete time steps in the range $(t - m, t]$ where each input $\boldsymbol{x}_i \in \mathbb{R}^d$. First, we aim to generate an "*event model*"[2] $\boldsymbol{z}^{(l)}$ which is a single bottleneck representation of the given inputs $\mathcal{X}^{(l)}$. To accomplish this, we define a function $f^{(l)} : \mathbb{R}^{m \times d} \mapsto \mathbb{R}^d$ with temporally shared learnable weights $\boldsymbol{\Phi}^{(l)}$ to evaluate the importance of each input in $\mathcal{X}^{(l)}$ for solving the prediction task at hand, as expressed in Equation (1).

$$\boldsymbol{z}^{(l)} = f^{(l)}(\mathcal{X}^{(l)}; \boldsymbol{\Phi}^{(l)}) \tag{1}$$

This event model will be trained to extract information from $\mathcal{X}^{(l)}$ that is helpful for hierarchical prediction. Ideally, the bottleneck representation should encode top-down semantics, which allow for event retrieval and a bottom-up interpretation of the input to minimize the prediction loss of the following input. The following subsection describes the learning objective to accomplish this encoding task.

## 2.2 Temporal prediction

At the core of our architecture is the prediction block, which serves two purposes: event demarcation and cross-layer communication. As previously mentioned, our architecture is built on the premise that minimizing the prediction loss is the only needed objective function for hierarchical event segmentation and representation learning.

**Cross-layer communication** allows the representation $\boldsymbol{z}^{(l)}$ to utilize information from higher $\{\boldsymbol{z}^{(l+1)}, \ldots, \boldsymbol{z}^{(L)}\}$ and lower layers $\{\boldsymbol{z}^{(1)}, \ldots, \boldsymbol{z}^{(l-1)}\}$ when predicting the next input at layer $l$,

---

[2]We use 'event model' and 'representation' interchangeably; terminologies defined in the appendix.

where $L$ is the total number of layers. Let $\mathcal{Z}_t = \{z_t^{(1)}, \dots, z_t^{(L)}\}$ be a set of event models where each element is the output of the temporal encoding function $f$ at its corresponding layer as expressed in Equation (1). Note that the same time variable $t$ is used for representation across layers for simplicity; however, each layer operates in its own subjective timescale. Let $p^{(l)} : \mathbb{R}^{L \times d} \mapsto \mathbb{R}^d$ be a function of $\mathcal{Z}$ to predict the next input at layer $l$ as expressed in Equation (2)

$$\hat{\boldsymbol{x}}_{t+1}^{(l)} = p^{(l)}(\mathcal{Z}_t; \boldsymbol{\Psi}^{(l)}) \tag{2}$$

where $\boldsymbol{\Psi}^{(l)}$ denotes the learnable parameters of the predictor at layer $l$. The difference between the layer's prediction $\hat{\boldsymbol{x}}_{t+1}^{(l)}$ and the actual input $\boldsymbol{x}_{t+1}^{(l)}$ is minimized, allowing the gradients to flow back into the $f^{(\cdot)}$ functions to modify each layer's representation as expressed in Equation (3).

$$\begin{aligned}
\mathcal{L}_{\text{pred}}(\hat{\boldsymbol{x}}_{t+1}^{(l)}, \boldsymbol{x}_{t+1}^{(l)}) &= \hat{\boldsymbol{x}}_{t+1}^{(l)} \sim \boldsymbol{x}_{t+1}^{(l)} \\
\boldsymbol{\Phi}^{*(i)}, \boldsymbol{\Psi}^{*(l)} &\leftarrow \underset{\boldsymbol{\Phi}^{(i)}, \boldsymbol{\Psi}^{(l)}}{\arg\min} \, \mathcal{L}_{\text{pred}}(\hat{\boldsymbol{x}}_{t+1}^{(l)}, \boldsymbol{x}_{t+1}^{(l)}) \qquad \forall \, i \in \{1 \dots L\}
\end{aligned} \tag{3}$$

The symbol $\sim$ represents an appropriate distance measure between two vectors.

**Event demarcation** is the process of detecting event boundaries by using the prediction loss, $\mathcal{L}_{\text{pred}}$, from Equation (3). As noted earlier, according to EST, when a boundary is detected, an event model transition occurs, and a new event model is used to explain the previously unpredictable observations. Instead of saving the event model to the event schemata at boundary locations as described in EST, we use it as a detached input (denoted by $\text{sg}[\cdot]$) to train the predictive model of the layer above it (*i.e.*, $\boldsymbol{x}^{(l+1)} \equiv \text{sg}[\boldsymbol{z}^{(l)}]$). We compute the running average of the prediction loss with a window of size $w$, expressed by Equation (5), and assume that a boundary is detected when the new prediction loss is higher than the smoothed prediction loss, as expressed by the decision function in Equation (4).

$$\delta(\boldsymbol{x}_t^{(l)}; \mu_{t-1}^{(l)}) = \begin{cases} 1 & \text{if } \mathcal{L}_{\text{pred}}(\hat{\boldsymbol{x}}_t^{(l)}, \boldsymbol{x}_t^{(l)}) > \mu_{t-1}^{(l)} \\ 0 & \text{otherwise} \end{cases} \tag{4}$$

where the running average is given by

$$\mu_t^{(l)} = \frac{1}{w} \sum_{i=t-w}^{t} \mathcal{L}_{\text{pred}}(\hat{\boldsymbol{x}}_i^{(l)}, \boldsymbol{x}_i^{(l)}) \tag{5}$$

## 2.3 Hierarchical gradient normalization

It is necessary to scale the gradients differently from conventional gradient updates because of the hierarchical nature of the model and its learning based on dynamic temporal contexts. There are three variables influencing the amount of accumulation of gradients:

1. The **relative timescale** between each layer is determined by the number of inputs. For instance, let the event encoder in layer $(l-2)$ have seen $a = |\mathcal{X}^{(l-2)}|$ inputs, that at layer $(l-1)$ have seen $b = |\mathcal{X}^{(l-1)}|$ inputs, and that at $l$ have seen $c = |\mathcal{X}^{(l)}|$ inputs. Then the input to layer $l$ is a result of seeing a total of $(a \cdot b \cdot c)$. This term can then be used to scale up the learning at any level $l$, expressed as $\prod_{i=1}^{l} |\mathcal{X}^{(i)}|$.

2. The **reach of influence** of each level's representation on a given level's encoder is influenced by its distance from another. For instance, if the input to $f^{(l)}$ comes from the levels $\{l+2, l+1, l, l-1, l-2\}$, then the weight of learning should be centered at $l$ and diminish as the distance increases. Such a weight at any level $l$ is given by $\alpha^{-|l-r|} \, \forall \, r \in \{-2, -1, 0, 1, 2\}$. To ensure that the learning values sum up to 1 when this scaling is applied, the weights are normalized to add up to 1 as $\frac{\alpha^{-|l-r|}}{\sum_{i=1}^{L} \alpha^{-|l-i|}}$.

3. The encoder receives **accumulated feedback** from predictors of all the layers; therefore the change in prediction loss with respect to encoder parameters in any given layer should be normalized by the total number of layers, given by $\frac{1}{L}$.

The temporal encoding model can be learned by scaling its gradients as expressed by the scaled Jacobian $\boldsymbol{J}'_{\mathcal{L}}(\boldsymbol{\Phi})$ in Equation (6).

$$\boldsymbol{J}'_{\mathcal{L}}(\boldsymbol{\Phi}) = \boldsymbol{C} \circ \boldsymbol{J}_{\mathcal{L}}(\boldsymbol{\Phi}) = \begin{bmatrix} c_{1,1} \frac{\partial \mathcal{L}^{(1)}}{\partial \boldsymbol{\Phi}^{(1)}} & \cdots & c_{1,L} \frac{\partial \mathcal{L}^{(1)}}{\partial \boldsymbol{\Phi}^{(L)}} \\ \vdots & \ddots & \vdots \\ c_{L,1} \frac{\partial \mathcal{L}^{(L)}}{\partial \boldsymbol{\Phi}^{(1)}} & \cdots & c_{L,L} \frac{\partial \mathcal{L}^{(L)}}{\partial \boldsymbol{\Phi}^{(L)}} \end{bmatrix} \tag{6}$$

where

$$c_{l,r} = \underbrace{\frac{1}{L}}_{\text{feedback}} \cdot \underbrace{\frac{\alpha^{-|l-r|}}{\sum_{i=1}^{L} \alpha^{-|l-i|}}}_{\text{reach of influence}} \cdot \underbrace{\prod_{i=1}^{l} |\mathcal{X}^{(i)}|}_{\text{timescale}} \tag{7}$$

Similarly, the temporal prediction model's gradients are controlled with scaling factors as expressed in Equation (8).

$$\boldsymbol{J}'_{\mathcal{L}}(\boldsymbol{\Psi}) = \boldsymbol{S} \circ \boldsymbol{J}_{\mathcal{L}}(\boldsymbol{\Psi}) = \begin{bmatrix} s_1 \frac{\partial \mathcal{L}^{(1)}}{\partial \boldsymbol{\Psi}^{(1)}} & \cdots & s_L \frac{\partial \mathcal{L}^{(L)}}{\partial \boldsymbol{\Psi}^{(L)}} \end{bmatrix} \tag{8}$$

where

$$s_l = \underbrace{\frac{1}{\sum_{i=1}^{L} \alpha^{-|l-i|}}}_{\text{reach of influence}} \cdot \underbrace{\prod_{i=1}^{l} |\mathcal{X}^{(i)}|}_{\text{timescale}} \tag{9}$$

## 3    Implementation

### 3.1    Training details

We resize video frames to $128 \times 128 \times 3$ and use a 4-layer CNN autoencoder (only for the first level) to project every frame to a single feature vector of dimension $1024$ for temporal processing. For predictive-based models (STREAMER and LSTM+AL), we sample frames at 2 fps, whereas for clustering-based models, we use a higher sampling rate (5 fps) to reduce noise during clustering. We use cosine similarity as the distance measure ($\sim$) and use the Adam optimizer with a constant learning rate of $1e-4$ for training. We do not use batch normalization, regularization (*i.e.*, dropout, weight decay), learning rate schedule, or data augmentation during training. We use transformer encoder architecture for functions $f$ and $p$; however, ablations show different architectural choices. A window size $w$ of 50 inputs (timescale respective) is used to compute the running average in Equation 5, and a new layer $(l+1)$ is added to the stack after layer $(l)$ has processed 50K inputs.

### 3.2    Delayed gradient stepping and distributed learning

Unlike our proposed approach, conventional deep learning networks do not utilize high-level outputs in the intermediate-level predictions. Since our model includes a top-down inference component, such that a lower level (*e.g.*, $(l)$) backpropagates its loss gradients into the temporal encoding functions of a higher level (*e.g.*, $f^{(>l)}$), we cannot apply the gradients immediately after loss calculation at layer $(l)$. Therefore, we allow for scaled (*i.e.*, Section 2.3 and Equation (8)) gradients to accumulate at all layers, then perform a single gradient step when the highest layer $L$ backpropagates its loss.

In our streaming hierarchical learning approach, event demarcation is based on the data (*i.e.*, some events are longer than others), posing a challenge for traditional parallelization schemes. We cannot directly batch events as inputs because each layer operates on a different subjective timeline. Therefore, each model is trained separately on a single stream of video data, and the models' parameters are averaged periodically during training. We train eight parallel streams on different sets of videos and average the models' parameters every 1K frames.

### 3.3    Datasets and comparisons

In our training and evaluation, we use two large-scale egocentric datasets: Ego4D [12] and EPIC-KITCHENS 100 [20]. Ego4D is a collection of videos with a total of 3670 hours of daily-life activity

Table 1: Event segmentation comparison of MoF and average IoU, evaluated on EPIC-KITCHENS. None of the methods listed below requires labels.[3] The column 'Layers' refers to the number of layers evaluated against the ground truth: 1 reports the performance of the best layer in the prediction hierarchy, whereas 3 uses the proposed Hierarchical Level Reduction algorithm for evaluation.

| Method | Backbone | Pretrained | Layers | Protocol 1 | | Protocol 2 | |
| --- | --- | --- | --- | --- | --- | --- | --- |
| | | | | MoF ↑ | IoU ↑ | MoF ↑ | IoU ↑ |
| LSTM+AL [1] | VGG16 [50] | ImageNet [46] | 1 | 0.694 | 0.417 | 0.659 | 0.442 |
| TW-FINCH [47] | | | 1 | 0.707 | 0.443 | 0.692 | 0.442 |
| Offline ABD [16] | MTRN [60] | EPIC 50 [12] | 1 | 0.704 | 0.438 | 0.699 | 0.432 |
| Online ABD [16] | | | 1 | 0.610 | 0.496 | 0.605 | 0.487 |
| STREAMER | 4-layer CNN | - | 1 | **0.759** | **0.508** | **0.754** | **0.489** |
| | | | 3 | **0.736** | **0.511** | **0.729** | **0.494** |

collected from 74 worldwide locations. EPIC-KITCHENS 100 contains 100 hours of egocentric video, 90K action segments, and 20K unique narrations. We train our model in a self-supervised layer-by-layer manner (using only RGB frames and inputting them exactly once) on a random 20% subset of Ego4D and 80% of EPIC-KITCHENS, then evaluate on the rest 20% of EPIC-KITCHENS. We define two evaluation protocols: Protocol 1 divides EPIC-KITCHENS such that the 20% test split comes from kitchens that have not been seen in the training set, whereas Protocol 2 ensures that the kitchens in the test set are also in the training set.

We compare our method with TW-FINCH [48], Offline ABD [16], Online ABD [16], and LSTM+AL [1]. ABD, to the best of our knowledge, is the state of the art in unsupervised event segmentation. Clustering-based event segmentation models do not evaluate on egocentric datasets due to the challenges of camera motion and noise. Most clustering based-approaches use pre-trained or optical-flow-based features, which are not effective when clustered in an egocentric setting. We re-implement ABD due to the unavailability of official code and use available official implementations for the other methods.

## 3.4 Evaluation metrics and protocols

**Event segmentation**    The 20K unique *narrations* in EPIC-KITCHENS include different labels referring to the same actions (*e.g.*, turn tap on, turn on tap); therefore we cannot evaluate the labeling performance of the model. We follow the protocol of LSTM+AL [1] to calculate the Jaccard index (IoU) and mean over frames (MoF) of the one-to-one mapping between the ground truth and predicted segments. Unlike LSTM+AL [1], which uses Hungarian matching to find the one-to-one mapping, we design a generalized recursive algorithm called Hierarchical Level Reduction (HLR) which finds a one-to-one mapping between the ground truth events and (a single-layer or multi-layer hierarchical) predicted events. A detailed explanation of the algorithm can be found in Supplementary Material.

**Representation quality**    To assess the quality of the learned representations, we use the large language model (LLM) GPT 3.5 to first create a dataset of events labels ranked by semantic similarity according to the LLM. In particular, we generate 1K data points sampled from EPIC-KITCHENS, where each data point comprises a 'query' narration and a set of 10 'key' narrations, and each key is ranked by its similarity to the query. We then retrieve the features for each event in the comparison and compute the appropriate vector similarity measure, and accordingly rank each key event. This rank list is then compared with the LLM ranking to report the MSE and the Levenshtein edit distance between them. Examples of LLM similarity rankings are available in Supplementary Material.

## 3.5 Experiments

We evaluate STREAMER's performance of event segmentation and compare it with streaming and clustering-based SOTA methods as shown in Table 1. Our findings show that the performance of a single layer in STREAMER's hierarchy (the best-performing layer out of three per video) and the

---

[3]Numbers in bold typeface indicate the highest performance; underline indicates the second highest

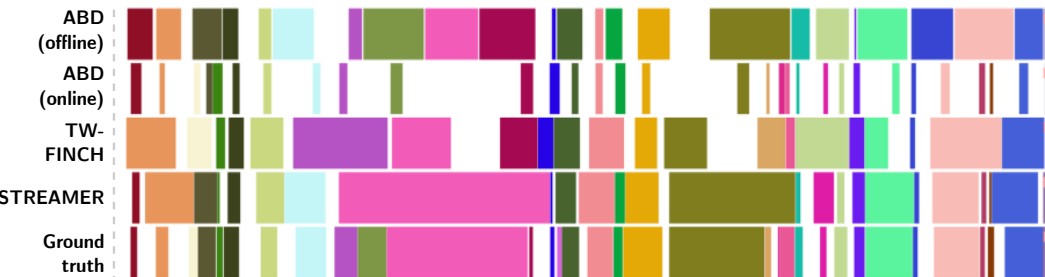

Figure 4: Qualitative comparisons of event segmentation. The Gantt chart shows a more accurate alignment of STREAMER's predictions with the ground truth compared to other methods.

full 3-layer hierarchy outperform all other state of the art using IoU and MoF metrics on both testing protocols. It is worth noting that all the other methods use a large CNN backbone with supervised pre-trained weights (some on the same test dataset: EPIC-KITCHENS), whereas our model is trained from scratch using random initialization with a simple 4-layer CNN. We show comparative qualitative results in Figure 4. More qualitative results are provided in Supplementary Material.

Additionally, we evaluate the quality of event representation in Table 2. We show that self-supervising STREAMER from randomly initialized weights outperforms most clustering-based approaches with pre-trained weights; we perform on par with TW-FINCH when using supervised EPIC-KITCHENS pre-trained weights. Qualitative results of retrieval for the first three nearest neighbors on all the events in the test split are shown in Figure 5. More qualitative results are reported in Supplementary Material.

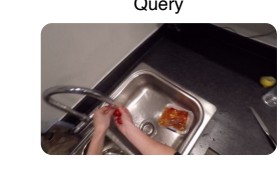

| Method | Weights | MSE ↓ | LD ↓ |
|---|---|---|---|
| **Supervised** | | | |
| TW-FINCH | EPIC | 1.00 | **0.67** |
| | IN | 1.018 | 0.710 |
| Offline ABD | EPIC | 1.02 | 0.71 |
| | IN | 1.005 | 0.708 |
| Online ABD | EPIC | 1.00 | 0.70 |
| | IN | 1.039 | 0.704 |
| **No supervison** | | | |
| STREAMER | - | **0.967** | 0.695 |

Table 2: Retrieval evaluation based on MSE and the Levenshtein edit distance (LD) of the features. All experiments are on EPIC-KITCHENS.[3]

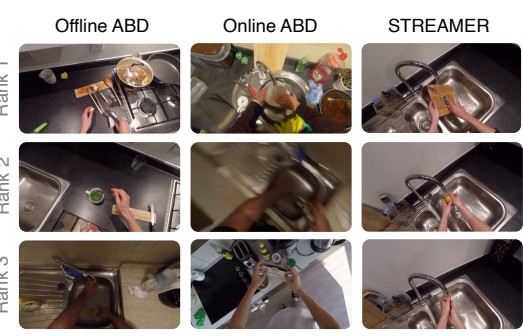

Figure 5: Qualitative examples of STREAMER's retrieval of relevant events compared to other methods.

**Ablations** We investigate three main aspects of STREAMER (Table 3): (1) varying the architecture of the temporal encoding model $f$, (2) varying the predictor function $p$, and (3) experimenting with the 'reach of influence' parameter $\alpha$ in Equation 7. Our findings suggest that STREAMER is robust to different architectural choices of $f$. Our experiments also illustrate the importance of the cross-layer communication of $p$: simply taking the average of $\mathcal{Z}$ as the prediction performs worse than applying a layer-specific MLP to the average; using a transformer to retrieve context from other layers dynamically performs the best. Finally, adjusting the reach of influence by gradient scaling improves the segmentation performance.

To determine the quality of the backbone features learned by STREAMER, we run ablations of using our 4-layer pretrained CNN features on SoTA clustering methods. The results, plotted in Figure 6, show significant improvement in the average mean over frames (MoF) performance of

event segmentation on the EPIC-KITCHENS dataset. This improvement could be attributed to the robust representations learned by the encoder through hierarchical predictive learning. In particular, since these features are learned through top-down optimization, the CNN backbone is able to predict longer events at higher levels, thus improving the features and contextualization quality.

| | MoF ↑ | IoU ↑ | MoF ↑ | IoU ↑ | MoF ↑ | IoU ↑ |
|---|---|---|---|---|---|---|
| $f$ | GRU | | LSTM | | Transformer | |
| **Best** | 0.759 | 0.503 | 0.761 | 0.506 | 0.759 | 0.508 |
| **HLR** | 0.740 | 0.503 | 0.737 | 0.502 | 0.736 | 0.511 |
| $p$ | Average | | Average + MLP | | Transformer | |
| **Best** | 0.742 | 0.479 | 0.749 | 0.493 | 0.759 | 0.508 |
| **HLR** | 0.725 | 0.486 | 0.728 | 0.494 | 0.736 | 0.511 |
| $\alpha$ | 1 | | 2 | | 3 | |
| **Best** | 0.756 | 0.493 | 0.797 | 0.498 | 0.759 | 0.508 |
| **HLR** | 0.737 | 0.498 | 0.732 | 0.497 | 0.736 | 0.511 |

Table 3: Ablations studies showing the model's MoF and IoU for different values of $\alpha$ (Equation (7)); different variants of the predictor $p$ and the temporal encoder $f$. 'Best' refers to the layer with the highest performance.

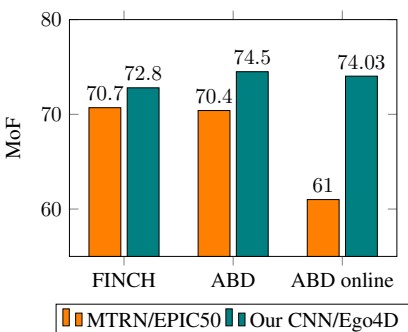

Figure 6: Performance increase of SoTA clustering-based methods when using STREAMER's pretrained 4-layer CNN features.

## 4 Conclusion

In conclusion, we present STREAMER, a self-supervised and structurally evolving hierarchical temporal segmentation model that is shown to perform well on egocentric videos and is robust to hyperparameter variations. The learned representations are hierarchical in nature, representing events at different levels of granularity and semantics. As part of this, we design a gradient scaling mechanism necessary for such hierarchical frameworks with varying time-scales.

STREAMER adheres to several biologically-inspired constraints and exhibits the ability to process long previous contexts in a streaming manner, seeing each input exactly once. Our method is designed to be trained in a streaming manner which allows models to perform inference simultaneously during training, appealing to applications that require real-time adaptability [32]. We demonstrate its ability to perform event segmentation on large egocentric videos of varying perceptual conditions and demonstrate the quality of the representations through event retrieval and similarity ranking experiments.

**Broader impact and limitations** STREAMER requires large amounts of data to model complex high-level causal structures, and the training time increases as a layer is added. However, as self-supervised learning is becoming of increasing essence, new models must be able to continually learn from large, unlabeled data from constantly evolving domains. STREAMER caters to such online learning paradigms by fully exploiting large unlabelled video data. A much broader impact of this method extends to multi-modal data and domains beyond egocentric videos.

## Acknowledgements

This research was supported by the US National Science Foundation Grants CNS 1513126 and IIS 1956050. The authors would like to thank Margrate Selwaness for her help with results visualizations.

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

# Supplementary Material

## A   Hierarchical level reduction

Since the ground truth annotations are provided as a single level of event annotations, it is not possible to compare them with rich hierarchical event segmentation predicted by STREAMER. For a given video and its ground truth annotations and the predicted annotations, several one-to-one mappings between them exist; we desire to find the one with the highest average IoU. In addition, it is necessary to ensure that the resulting one-to-one mapping does not contain temporally overlapping predicted annotations.

To solve this optimization problem, we design a hierarchical level reduction (HLR) algorithm that reduces multiple layers of hierarchical events down to a single layer by selecting prediction events that maximize IoU with the ground truth while ensuring no overlap of events during reduction. We design HLR as a recursive greedy optimization strategy. At each recursive level of Algorithm 1, multiple ground truth events are competing to be assigned to a single predicted event, so HLR returns the best of the two options (line 18): (1) assigning the event to the ground truth with the highest IoU, and (2) averaging the outputs of same recursive function over all children of the predicted event.

---

**Algorithm 1** : **Hierarchy Level Reduction**. Given a list of the highest level annotations $\mathcal{A}^L$ from the predicted hierarchy and the ground truth annotations $\mathcal{G}$, this algorithm finds the optimal match of the predicted annotations across the hierarchy with the ground truth while avoiding any temporal overlap between events.

**Input:**
   $\mathcal{A}^L$: a list of predicted events at the highest level $L$
   $\mathcal{G}$: a list of ground truth event annotations
**Output:**
   The overall IoU of the resulting hierarchy reduction

```
 1:  procedure FINDMATCHES(𝒜ˡ, 𝒢)
 2:      for all g ∈ 𝒢 do
 3:          max_ious ← {IoU(g, a) | ∀ a ∈ 𝒜ˡ}
 4:          a* ← arg max (max_ious)
 5:          a*.matches.push(g)
 6:          a*.ious.push(ious[a*])
 7:      end for
 8:      for all a ∈ 𝒜ˡ do
 9:          FINDMATCHES(a.children, a.matches)
10:      end for
11:  end procedure

12:  procedure REDUCELEVELS(a)
13:      if |a.matches| = 1 then return a.ious[0]
14:      end if
15:      h ← max(a.ious)
16:      if |a.children| = 0 then return h
17:      else
18:          return max(h, mean({REDUCELEVELS(c)) | ∀ c ∈ a.children})
19:      end if
20:  end procedure

21:  FINDMATCHES(𝒜ᴸ, 𝒢)
22:  return mean({REDUCELEVELS(a) | ∀ a ∈ 𝒜ᴸ})
```

---

For models generating a hierarchical structure of events, the proposed Hierarchical Level Reduction (HLR) algorithm could be applied for comparison and evaluation. On the other hand, methods that generate a single layer of event segments can directly compare to our 1-layer evaluation reported in Table 1.

## B  Qualitative results

This section contains more qualitative results of STEAMER. A main argument of our paper is that the ground truth annotations (narrations) of events in EPIC-KITCHENS are not consistent and are sometimes redundant. Figure 7 illustrates one such case. Figure 8 shows an example of STREAMER's hierarchical annotations. We refer the reader to the supplementary video for more examples.

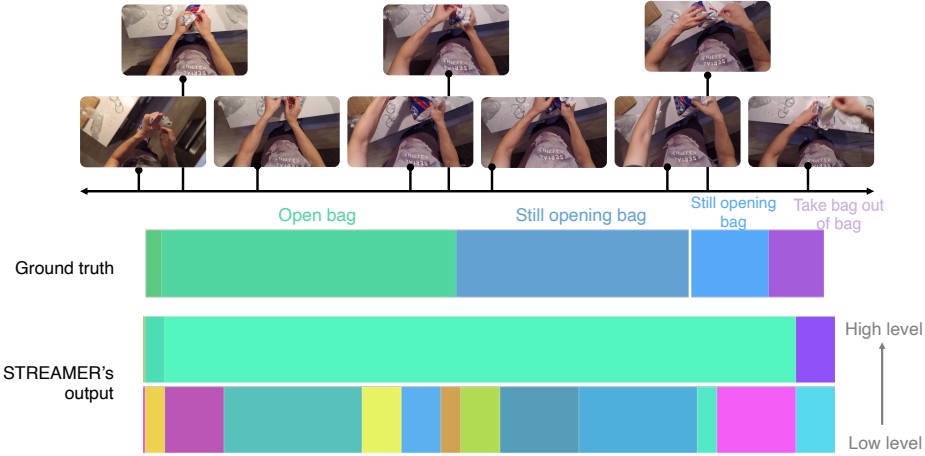

Figure 7: This figure illustrates the effect of inconsistent ground truth on the model's evaluation performance. In this segment of a video from EPIC-KITCHENS, the ground truth consists of the same narration annotated thrice in succession (*open bag* ▪, *still opening bag* ▪, *still opening bag* ▪). Although our model could correctly detect this narration to its entirety (the middle row ▪), its IoU is low, thus affecting its overall evaluation score. Such inconsistencies and redundancies are prevalent throughout the dataset.

Given a video snippet of an event (which we will refer to as a 'query'), can the model retrieve semantically similar video snippets from across the dataset? To determine this, we perform event retrieval by representation: we first generate a representation for a random query which is then compared with the representations of events from all the videos in the dataset. Based on an appropriate similarity measure as required by the model, we select the top-few nearest matches and qualitatively examine the result.

Figure 9 shows an example of the top-three similar matches compared with ABD. Figure 10 shows more examples of STREAMER's event retrieval, displaying the best of the top three matches. Distance in feature space is calculated by the cosine similarity for our method and the Euclidean distance for ABD.

## C  Retrieval: quantitative analysis

In addition to the qualitative results, we perform more quantitative experiments on retrieval. As described in the main text of this work, we use the large language model (LLM) GPT 3.5 to create a dataset of event labels from EPIC-KITCHENS ranked by the semantic similarity. The dataset contains 1K comparisons where each comparison comprises a 'query' narration and a set of 10 'key' narrations, and each key is ranked by its similarity to the query according to the LLM. The keys are ranked according to the distance of their representations in the feature space. The two rank lists are compared based on 1) Mean Squared Error (MSE) and 2) the Levenshtein edit distance. Listing 1

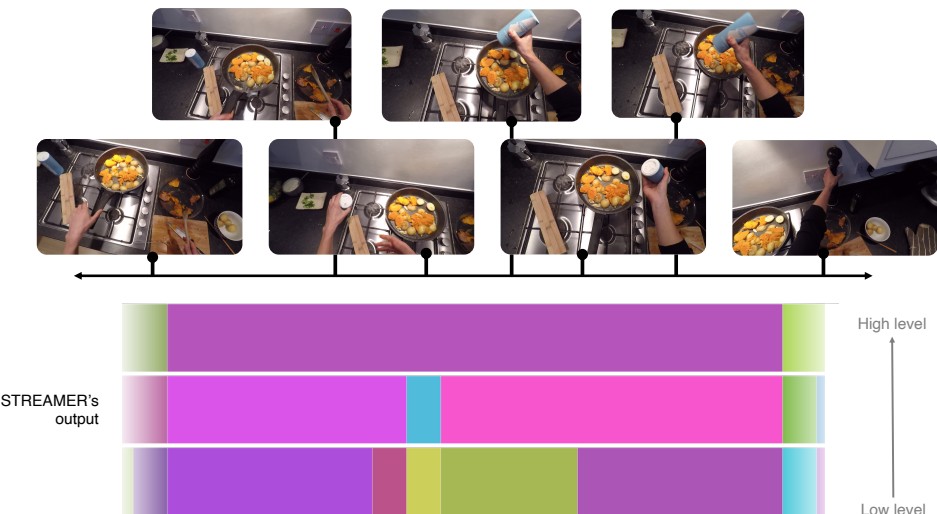

Figure 8: Given a sequence of temporal perceptual inputs (*e.g.*, video), our model learns to represent them at varying levels of detail. This figure illustrates the predictions made by our model on a video from EPIC-KITCHENS at three levels: the highest level (the top row in the bar chart) captures a high-level, low-detail concept (seasoning vegetables ■); the middle row captures events at relatively finer detail (mixing vegetables ■ and adding salt ■); and the last row captures the events in much more granular detail. Video available as supplementary.

shows the prompt used to generate the dataset and Table 5 shows some examples of LLM similarity rankings in the created dataset.

```
1    prompt = f"""
2    Given a list of phrase pairs, compute the semantic similarity
     ↪ between the phrases in each pair and rank in the continuous
     ↪ range of 0 to 10 where 10 is most similar.
3
4    The list is: {queries}
5
6    Just return a list of decimal numbers. No explanation.\n"""
```

Listing 1: The LLM prompt used to generate the dataset for retrieval quality evaluation (ranks divided by 10 to be in $(0, 1)$)

## D    Implementation details

Let $k$ be the total number of narrations in the ground truth for a given video plus one (for the background).

**TW-FINCH, LSTM-AL**    The official implementations of TW-FINCH and LSTM-AL are available on GitHub (TW-FINCH, LSTM-AL). We use the provided code to run our comparisons.

For LSTM-AL, the required number of clusters for each video being clustered was set to $k$. For LSTM-AL, the order of peak detection was set to 2 frames (sampled at 2 fps) to optimize the best results.

**ABD**    ABD, both offline and online clustering versions, had to be re-implemented based on the implementation details of the paper. For offline clustering, the window size was set to 5, the order of non-max suppression to 10, and the average number of actions to $k$. For the online clustering variant, the window size was set to 15, the order of non-max suppression to 40, and the lower quantile to 0.25 as prescribed by the paper.

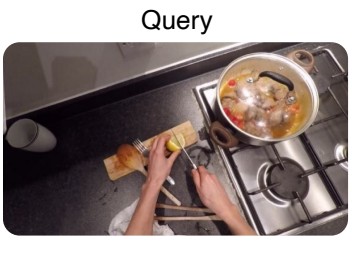

Query

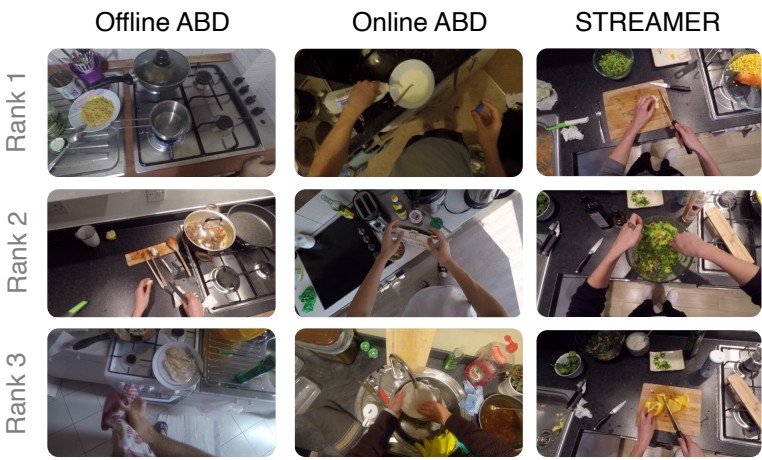

Offline ABD        Online ABD        STREAMER

Rank 1    Rank 2    Rank 3

Figure 9: A comparison of top-three nearest neighbors retrievals with ABD.

# E   Literature survey

In this section we provide a more detailed review of related works. We start by discussing single-layered event segmentation and boundary detection works, followed by a summary of existing ideas, inspirations and implementations of hierarchical models. We end our survey by relating to successful prediction-based models for representation learning.

## E.1   Event segmentation

**Supervised event segmentation**   One effective approach to event segmentation is to label every frame with a class then fully supervise parameterized learning models (*e.g.*, neural networks) to classify each frame. This eventually groups frames into events; however, a major drawback of these methods is the cost of fine-grained frame annotations. Additionally, fully supervised methods fail to generalize well due to the labels being in a closed set. Different model variations and approaches have been tested, such as using an encoder-decoder temporal convolutional network (ED-TCN) [35], multi-stage temporal convolution network (MS-TCN) [17], or a spatiotemporal CNN model [36]. HASR [2] refines the output predictions of existing segmentation models by utilizing segment-level representations, whereas ASRF [30] improves the segmentation performances by regressing the action boundary probability using representations with a wide temporal receptive field.

**Weakly-supervised event segmentation**   In order to avoid the costly process of direct frame labeling, researchers have developed weakly-supervised methods that utilize metadata (*e.g.*, captions or narrations) to guide the training process instead of relying on explicit training labels. These approaches have been explored in various studies [45, 4, 15, 29, 3], aiming to reduce the dependency on labeled data. However, one limitation is that the required metadata may not always present in the dataset, which restricts their applicability to many real-world scenarios.

**Unsupervised event segmentation**   To fully eliminate the need for labeling, some approaches [48, 16] attempt to cluster high-level features of frames, which also eliminates training. The segmentation

| Query | Best match | | Query | Best match |

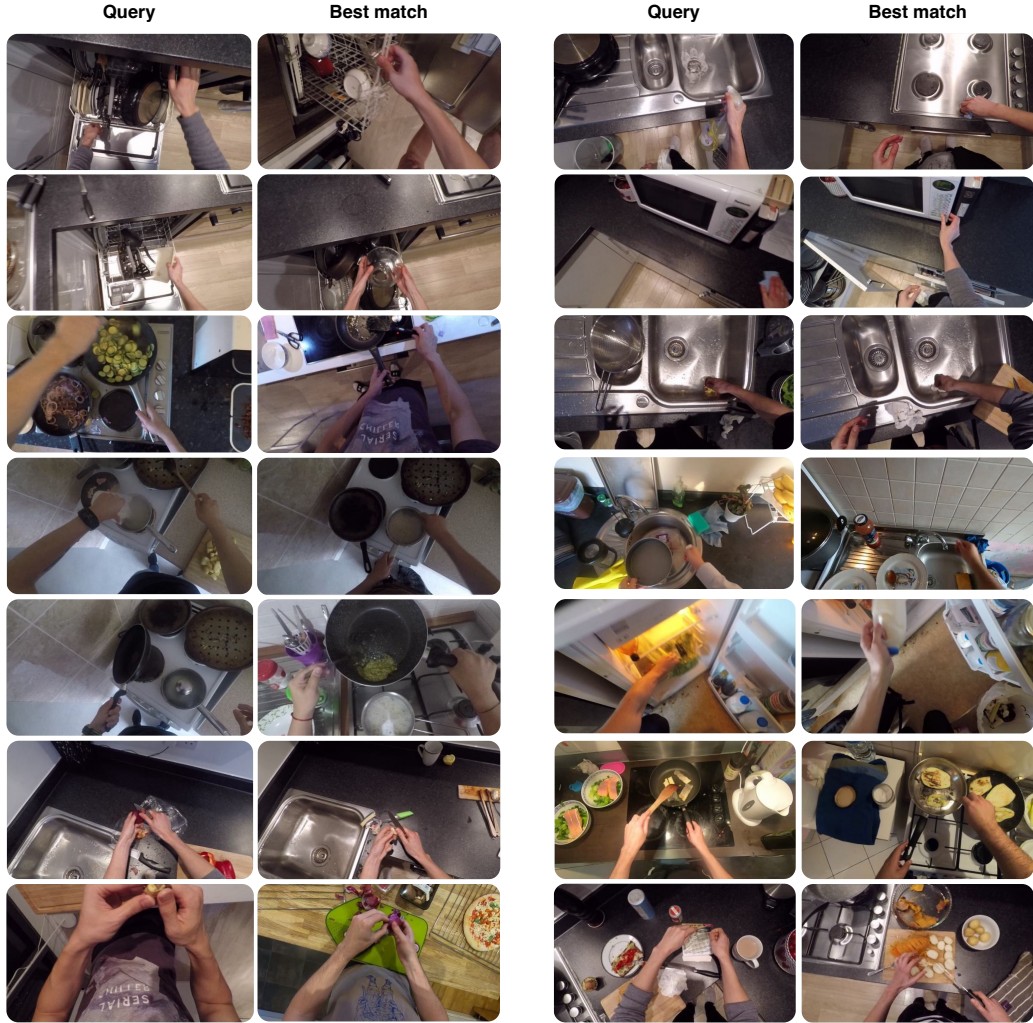

Figure 10: Random queries and the corresponding best matches, chosen from a set of top-three candidates for each query on EPIC-KITCHENS.

performance is directly proportional to the quality of features used in clustering. Therefore when using optical flow-based features, performance will suffer if applied to moving camera videos (*e.g.*, egocentric). Note that in our comparisons we provide these approaches with EPIC-KITCHENS supervised features to enhance their performance. Other methods, such as CTE [33] and TOT [34] utilize the order of actions and decode consistent labels using the Viterbi algorithm.

**Event boundary detection** Other works (more similar to ours) have formulated the segmentation problem as a boundary detection problem. Generic Event Boundary Detection (GEBD [49]) proposed to use the difference in context embedding before and after each frame to detect boundaries as peaks. Other works [1, 55] use inspiration from [59] to detect boundaries as peaks in the prediction error signal. These approaches use only low-order prediction (based on pretrained supervised high-level features) to form a single layer of event representation and fail to perform higher-order predictions.

## E.2 Hierarchical structure modeling

**Chunking** Recent work [56] provide a way to chunk data to learn a hierarchical structure by explicitly forming a dictionary of patterns at each level. These methods can be challenging to scale up to real data due to the enormous number of event possibilities to be stored, thus demonstrated only on toy dataset examples of text and vision. Additionally, quantized inputs are required to form these

patterns. In contrast, STREAMER does not save patterns but uses learnable networks to encode and predict future inputs. We demonstrate our results on challenging egocentric data.

**GLOM and H-JEPA**   Hinton [26] proposed an idea paper for "GLOM", which attempts to build a part-whole hierarchy by assuming a cortical column for each patch in an image, where each layer in every column receives contributions from nearby columns at higher and lower levels. The hierarchy, or parse tree, is formed by forcing consensus laterally between each level, of each column, using a similarity function. Although we share the same motivation with GLOM, we significantly differ in execution. Instead of using a similarity function to force forming "islands of agreement", STREAMER relies on prediction error to discover events and explicitly send their representation to higher level in a dynamically evolving structure (*i.e.*, layer-by-layer). Perhaps we share more algorithmic similarity with LeCun's [37] idea paper, "H-JEPA". The paper proposes to use a hierarchical joint embedding predictive architecture which uses self-supervised predictive learning to predict at different timescales at various levels in the hierarchy. This is also supported by evidence from neuroscience [8] suggesting the importance of long-range forecast representations in improving brain-mapping functionality. H-JEPA suggests that higher levels predict further into the future than lower levels; however, it does not provide a working algorithm for segmentation or cross-layer communication. It is to be noted that both works by Hinton and LeCun only provide ideas and not algorithms/results.

### E.3   Predictive modeling

**Natural language processing**   Self-supervised prediction of inputs has shown tremendous success in the field of NLP, specifically masked language models. Recent large language models [13, 5, 52, 54] are based on the core idea of using context to predict missing inputs. The guiding principle of training these models is that successfully predicting missing inputs implies encoding good representation of the context.

**Computer vision**   A similar principle is used to train image representation learning models. Masked Auto Encoder (MAE [25]) is trained to predict missing image patches using a transformer architecture, thus performing better on downstream tasks using the enhanced representations. A family of representation learning methods [21, 6, 9, 7, 43] learn useful representations by predicting augmented views of the input. These augmented views attempt to simulate real-world augmentations (*e.g.*, cropping simulates a moving camera), therefore these methods may also be viewed as predicting future frames in a video sequence. Similar ideas can also be seen applied to video representations learning. PredNet [38] attempts to implement cognitively-inspired predictive coding theories for future frame prediction. Other methods [22, 23] use dense predictive coding to learn video representations for retrieval and fine-tuning on downstream tasks. Unlike STREAMER, these methods do not generalize to learning hierarchical representations using predictive learning.

### E.4   Relevant datasets

STREAMER is a self-supervised architecture that relies on predictive learning for hierarchical segmentation. In our model, higher-order predictive layers receive sparser learning signals than lower-order layers, because the first layer directly predicts frames, whereas higher layers only receive events that cannot be predicted at lower levels. Short videos do not allow for higher order predictions and learning long-term temporal dependencies. Therefore, training higher levels in the hierarchy requires a large dataset (*i.e.*, total number of hours) and longer videos (*i.e.*, average video duration) in order to model high-level events. These requirements constraint the choice of datasets on which we can run and evaluate STREAMER.

Based on our review of available datasets for both egocentric and exocentric settings, as shown in the Table 4, many of the available datasets, typically used in event segmentation, do not provide long enough videos. MovieNet and NewsNet are two large datasets with long videos but have not yet been released to the public. In addition, MovieNet does not contain action segments; it contains coarse scene segments. The only available options to train and evaluate STREAMER is large-scale egocentric datasets, where the available datasets provide large enough scale (*i.e.*, total number of hours) with a long average duration per video for streaming and high-order temporal prediction.

Table 4: Various egocentric and exocentric datasets with total hours of recording and average duration statistics. * Not released as of this writing.

| | Dataset | Total Hours | Avg. Duration (min) | Large Datasets | Long Videos |
|---|---|---|---|---|---|
| Egocentric | Ego4D | 3670 | 23 | ✓ | ✓ |
| | EPIC 55 | 55 | 10.5 | ✓ | ✓ |
| | EPIC 100 | 100 | 8.5 | ✓ | ✓ |
| | GTEA | 0.58 | 1.23 | ✗ | ✗ |
| Exocentric | Breakfast Action | 77 | 2.3 | ✓ | ✗ |
| | YouTube Instructional | 5 | 2 | ✗ | ✗ |
| | Hollywood Extended | 3.7 | 0.23 | ✗ | ✗ |
| | 50 Salads | 4.5 | 4.8 | ✗ | ✗ |
| | FineGym | 708 | 0.91 | ✓ | ✗ |
| | ActivityNet Captions | 849 | 2.5 | ✓ | ✗ |
| | MovieNet* | 2174 | 117 | ✓ | ✓ |
| | NewsNet* | 946 | 57 | ✓ | ✓ |

# F   Glossaries

- **Contextualized inference** In this work, *contextualized inference* refers to the ability of the model to predict a future event by using contextual representations at various levels of the event hierarchy.

- **Contextualized optimization** *Contextualized optimization* refers to the ability of a layer to optimize the representations of other layers through its own prediction loss.

- **Event** An *event* is defined as "a segment in time that is perceived by an observer to have a beginning and an end" [59]

- **Event model** In cognitive psychology literature, an *event model* is defined as "a representation of what is happening now, which is robust to transient variability in the sensory input" [59]; in this work, we use 'event model' and 'representation' interchangeably.

- **Event demarcation, event segmentation** *Event demarcation* is the process of detecting event boundaries by using the prediction loss, whereas *event segmentation* is the task of segmenting videos (or sensory inputs) into meaningful events. In other words, event segmentation is the goal, and event demarcation is one way to achieve it: event segmentation could be performed in other ways, such as labeling each frame in a supervised framework.

- **Predictive learning** *Predictive learning* refers to the brain's ability to generate predictions about future events based on past experiences.

- **Segmentation boundary** The end of an event and the beginning of the next is a *segmentation boundary*, marking an event transition.

Table 5: This table shows some examples of the similarity of narrations of 'key events' to a 'query event' as determined by GPT 3.5 (`text-davinci-003`). A 1 scoring means most similar, and a 0 least.

| Query | Keys | | | | | | | | | |
|---|---|---|---|---|---|---|---|---|---|---|
| open bottle | stop processor | open freezer | enter kitchen | open door | pick up bowl | open fridge | clean kitchen counter | move cheese | put down sausage | put down pan |
| | 0.25 | 0.8 | 0.5 | 0.7 | 0.4 | 0.85 | 0.45 | 0.3 | 0.45 | 0.4 |
| open drawer | pick up stone | check chicken | still cleaning chopping board | shake off courgette | rinse pot | pick up | close dishwasher | take out pasta | still scoop the kiwi | pick up something |
| | 0.75 | 0.55 | 0.65 | 0.65 | 0.7 | 0.8 | 0.7 | 0.85 | 0.65 | 0.8 |
| wash two leaves | wash knife | put down pepper | rinse pan | put away spoon | wash lid | grab bag | get kettle | pour olive oil in pan | stir courgette | pick up forks |
| | 0.75 | 0.45 | 0.85 | 0.65 | 0.95 | 0.35 | 0.45 | 0.75 | 0.65 | 0.55 |
| wash pot | wash pan | open jar | open cupboard | pick up mug | pick up aubergine | pour milk into cereal bowl | rinse cloth | lay aubergine | close fridge | open cupboards |
| | 0.85 | 0.65 | 0.75 | 0.8 | 0.85 | 0.9 | 0.85 | 0.8 | 0.75 | 0.75 |
| pick up bowl | turn off tap | put down salt | cut tomato | stir onion | pick up bowl | stir chicken | place pan | grab salt container | open cupboard | pour milk into glass |
| | 0.75 | 0.55 | 0.45 | 0.65 | 1.0 | 0.7 | 0.6 | 0.65 | 0.5 | 0.65 |
| get chopping board | move chair | open washing machine door | rinse hands | check oil | pour detergent | rinse hands | check temperature | put bread onto tray | wash chopping board with sponge | select schedule |
| | 0.25 | 0.5 | 0.8 | 0.3 | 0.5 | 0.8 | 0.6 | 0.7 | 0.9 | 0.4 |
| put pot | turn on tap | clean cooker | still rinsing spatula | put colander on pot | put fork in bowl | put lid on pot | put lid on sun-dried tomatoes | rinse hands | wipe down counter | move mouse |
| | 0.75 | 0.45 | 0.55 | 0.85 | 0.65 | 0.95 | 0.45 | 0.55 | 0.65 | 0.45 |
| open drawer | take box | pick up jar | close drawer | rinse mug | close hob cover | scrape tomato | lift plate | wash knife | get plastic trash bag | check timer |
| | 0.75 | 0.85 | 0.95 | 0.65 | 0.75 | 0.65 | 0.75 | 0.85 | 0.65 | 0.85 |
| close jug | shake coffee maker | pick up plate | take napkin | get weighing | still taking skin off meat with knife | pick up lid | close tap | get chopping board | put down spoon | put down bowl |
| | 0.25 | 0.5 | 0.5 | 0.25 | 0.05 | 0.75 | 0.9 | 0.5 | 0.75 | 0.75 |
| wash bowl | wash coffee pot | put down knife | put down detergent | throw away onion skin | turn on tap | rinse chopping block | mix nuts with oats | dry hands | place sponge away | open tap |
| | 0.95 | 0.65 | 0.85 | 0.45 | 0.95 | 0.85 | 0.65 | 0.95 | 0.85 | 0.95 |