# OpenReview forum: "STREAMER: Streaming Representation Learning and Event Segmentation in a Hierarchical Manner"
_NeurIPS.cc/2023/Conference — NeurIPS 2023 poster_

### Official Review · Reviewer_s9dg · 2023-07-07

**Soundness:** 3 good
**Presentation:** 2 fair
**Contribution:** 2 fair
**Rating:** 4
**Confidence:** 3

**Summary:**

This paper introduces STREAMER, a self-supervised architecture for hierarchical representation learning and segmentation of streaming perceptual inputs. The model adapts to the complexity of the input domain through layer-by-layer training, aiming to accurately predict future events while providing necessary information to other levels. By detecting prediction error peaks, an event hierarchy is constructed, enabling a bottom-up information flow. The model is fully self-supervised, trained in a streaming manner, and demonstrates promising performance on temporal event segmentation and event retrieval tasks.

**Strengths:**

Most parts of this paper are well written which demonstrates its methodology and experiments. Especially, the method of this paper is easy to follow and the provided visualization is a plus to understand the proposed algorithm.

The idea of leveraging the global context is inspiring, which benefits to segment the events. In addition, the self-supervised learning pipeline significantly reduces the efforts in the annotation.

The experimental results and visualizations look promising on the Epic-Kitchen dataset.


**Weaknesses:**

The terminology used in this paper is confusing, for example, event model vs. representation, event demarcation vs. event segmentation.

The motivation of using streaming manner is not well justified in this work. It will be great to elaborate this part.

In Table 1, this work only reports the experimental results on EPIC-KITCHEN dataset, and there are few numbers of previous work comparisons. This is hard to justify the effectiveness of the proposed method.


**Questions:**

How to deal with the non-action clips in the untrimmed video sequences?

How about the computational cost of the proposed method?

**Limitations:**

as is mentioned in the weakness.

---

> ### Author Rebuttal · Authors · 2023-08-08
>
> ### Weaknesses:
>
> 1.  We will clarify the terminologies in the camera ready submission. For clarification:
>     - *Event model* and *representation*: As defined in cognitive psychology literature, "An event model is a representation of what is happening now, which is robust to transient variability in the sensory input" [1]; as noted in Footnote 2 (Page 5), we use 'event model' and 'representation' interchangeably.
>     - *Event demarcation* and *event segmentation*. As defined in L181, "Event demarcation is the process of detecting event boundaries by using the prediction loss" whereas "event segmentation" is the task of segmenting videos (or sensory inputs) into meaning events, as is often understood in literature. Event segmentation is the goal, and event demarcation is one way to achieve it. Event segmentation could be performed in other ways, such as labelling each frame in a supervised framework.
> 2. There are several motivations to learning in a streaming manner, which we will include in the camera ready submission, such as:
>     - Models trained in a streaming manner are more biologically plausible, as mentioned in L47, since the learning in the neocortex occurs continuously.
>     - Stream learning allows models to perform inference simultaneously during training which appeals to application requiring real-time adaptability
>     - When using stream learning, the model processes each input (i.e., frame) only once and does not store it to for additional training. Therefore, there are no significant storage requirements.
>     - The training can be scaled up to possibly process infinite inputs, which is not possible with conventional epoch-based training.
>     - Such setting enables learning from evolving data streams.
> 3. To the best of our knowledge, we are the first to perform hierarchical event segmentation on EPIC-KITCHENS, therefore the available methods for comparison are very limited. In fact, we had to re-run/re-implement the SOTA approaches. Furthermore, building higher-level hierarchical events in a self-supervised manner requires long videos. higher-order prediction tasks receive sparser learning signals than lower-order predictions (i.e., the first layer directly predicts frames, whereas higher layers only receive events that cannot be predicted at lower levels). Therefore, training higher levels in the hierarchy requires a large dataset (i.e., total number of hours) and longer videos (i.e., average video duration) in order to model high-level events (see figure to training time requirements in global response). We have included a table in the global response summarizing the statistics of relevant datasets suggesting lack of available long-video datasets for stream hierarchical learning.
>
> ### Questions:
>
> 1. Our approach works on non-actions and actions clips similarly. If the event is repeatable and predictable (e.g., moving across the kitchen), then lower level event models will be able to predict all frames in the non-action clips and segment them from other action clips. Otherwise, boundaries will exist at high prediction errors.
> 2. STREAMER processes frames at 3.3 frames per second (simultaneous training and inference); however, our predictive approach does not require processing consecutive frames. In practice, we sample 2 frames per second of the input 60fps videos, allowing STREAMER to process video at a speed of 90.9 fps. The results in the experimental section are shown with stream processing at 90.9 fps.
>
> [1] Jeffrey M Zacks, Nicole K Speer, Khena M Swallow, Todd S Braver, and Jeremy R Reynolds. Event perception: a mind-brain perspective. Psychological bulletin, 133(2):273, 2007
>
> *We hope to have addressed all your questions, and look forward to any further discussions*

---

### Official Review · Reviewer_egXC · 2023-07-07

**Soundness:** 3 good
**Presentation:** 3 good
**Contribution:** 3 good
**Rating:** 5
**Confidence:** 4

**Summary:**

This paper tackles the task of event segmentation in videos, with a particular emphasis on trying to perform this task in a self-supervised setting. The authors draw inspiration from event segmentation theories from human perception to design a model, STREAMER, that semantically groups inputs at varying lengths into events (in a streaming manner), that can then be hierarchically clustered to obtain desired segmentations (evaluation is done by greedy matching high IoU overlap with groundtruth segments). The modeling approach offers a different style from standard techniques, with bottom-up and top-down optimization that encourages a natural hierarchical segmentation. The learning is also done in a streaming manner, with a single pass over the input dataset (and multiple parallel "streams" with a federated learning style). The authors train their proposed model on a combined subset of Ego4D and EPIC Kitchens data, and evaluate on a subset of EPIC Kitchens event segmentation (both in settings with seen and unseen kitchens).

**Strengths:**

`+` The task of video event segmentation is important for video analysis and understanding in general.

`+` Architecture designs inspired by human perception and event segmentation theory are generally interesting directions to push on, even if not necessarily in the mainstream.

`+` The proposed technique, while it borrows aspects from different prior works on event segmentation and dense-predictive coding from self-supervised learning, feels interesting.

`+` The evaluation in a challenging egocentric setting is also interesting and may provide insights for downstream applications to robotics, etc.



**Weaknesses:**

`-` The work seems to have a limited evaluation overall, compared to other recent work in video segmentation with SSL. For example, the key relevant prior method (ABD) performs fast unsupervised action boundary detection and they measure performance across a range of datasets from youtube instructional videos to hollywood extended (which also has a lot of background frames). In contrast, the model here focuses exclusively on the egocentric setting, with evaluation only on EPIC Kitchens, which means it is unclear how the technique generalizes to other large scale video settings.

`-` Relatedly, the discussion of related work in the main paper (and to some extent, the supplement), seems to be limited regarding both the state-of-the-art in video segmentation overall (including more supervised settings with hierarchical event segmentation, like [M1]), and more classical video segmentation work (like [M2]). Having this kind of discussion would help to improve the level that the paper is self-contained, for general readers who may be less familiar with this space.

example missing references:
[M1] "Refining Action Segmentation with Hierarchical Video Representations", ICCV 2021
[M2] "Streaming Hierarchical Video Segmentation based on an Observation Scale", ICIP 2014


**Questions:**

Overall, the preliminary rating for this paper is borderline (leaning initially positive, since I think it is good to encourage different architecture settings / reference points, and the egocentric setting is good to see as well, but overall still borderline). The key points to address for the rebuttal are in the weaknesses above. In addition, there are also a few more clarifying questions:
1. Why does Protocol 1 have the same or better performance to Protocol 2, when Protocol 2 is the one that ensures the same kitchens from the train set are considered in the test? (per L247-248 and Table 1)
2. What does 1 layer mean in Table 1 for this proposed method? Does this mean there is no inter-layer bottom-up/top-down message passing, as described in the technical approach? There doesn't seem to be a significant difference between 1 and 3 layers (except some tradeoff on the metrics).
3. Also in Table 1, how does the EPIC 50 pretraining + MRTN backbone compare with that done here? (given the larger Ego4D training, is it possible that this alone accounts for differences?)
4. If the method needs this specific egocentric setting to work (and doesn't generalize to other settings well) - what are the reasons behind such a limitation with this approach?

As an additional note, the qualitative results in the supplement are interesting, especially the cases where the EPIC annotations make better sense to merge.

**Limitations:**

The authors provide a discussion of limitations in the end of the paper (though I do wonder how much the layer-wise restriction applies, if they are only using 1 layer?). The societal impact seems similar to other video modeling work.

---

> ### Author Rebuttal · Authors · 2023-08-08
>
> Thank you for your remarks on the qualitative results and highlighting the strengths of our idea.
>
> ### Weaknesses:
>
> 1. Building higher-level hierarchical events in a self-supervised manner requires long videos. Existing unsupervised methods, like ABD, are clustering-based methods which do not require training because they rely on precomputed features, therefore they can be applied to videos of any lengths. In contrast, our method learns its features from scratch. In addition, higher-order prediction tasks receive sparser learning signals than lower-order predictions (i.e., the first layer directly predicts frames, whereas higher layers only receive events that cannot be predicted at lower levels). Therefore, training higher levels in the hierarchy requires a large dataset (i.e., total number of hours) and longer videos (i.e., average video duration) in order to model high-level events (see figure to training time requirements in global response). We have included a table in the global response summarizing the statistics of relevant datasets.
> 2. Thank you for the suggestions, we will included the provided references and a more detailed related work section in the camera ready submission.
>
> ### Questions:
> 1. These protocols help determine if the model has overfit on specific visual features (such as kitchens, lighting) or if it is capable of modeling the robust event representations instead. The performance numbers of STEAMER show that the model is learning more robust representations of events instead of learning kitchen specific actions. It seems that the model is not overfitting to specific kitchen settings, one reason could be that since the higher levels are trained to embed more robust general understanding of events (low-detail), they regularize the performance of lower level events during prediction (through top-down inference) leading to on par performance between the two evaluation protocols. Also, as you noted, some EPIC-KITCHENS groundtruth annotations are better merged, therefore, introducing inconsistencies between the protocols' test sets.
> 2. Thanks for pointing this out. We will clarify it in the camera ready submission. We use two types of evaluation; “1 layer” denotes choosing a single layer out of the 3-layer hierarchy that maximizes the IoU with groudtruth annotations, whereas “3 layers” denote using the proposed Hierarchical Level Reduction algorithm (shown in the supplementary material). The small difference between both evaluation types validates that most groundtruth annotations come from a single level in our hierarchical output, which shows the consistency in our event level predictions. However, our model still outputs useful subevents and parent events to model a full hierarchy beyond what is provided in the groundtruth annotations.
> 3. We do not believe that pretraining with a 4-layer CNN backbone on a subset of out-of-domain dataset (i.e., ego4D) can by itself outperform supervised features on in-domain dataset (i.e., EPIC-KITCHENS) with large backbone. The LSTM+AL [1] method has been also trained on Ego4D and we still outperform it showing the efficacy of our hierarchical approach.
> 4. Our method does not have specific constraints on the type of input data it can work on; however, as mentioned in [weakness 1] above, our approach requires long ego/exo-centric for modeling higher level events. In our review of other datasets, we could not find datasets with long enough average durations outside of the egocentric domain. We have included a table in the global response summarizing the statistics of relevant datasets.
>
> ### Limitation:
> Just to clarify: as mentioned in [Question 2] above, "1-layer" refers to evaluating the model's output by the single best layer from the hierarchy.
>
> [1] Aakur, Sathyanarayanan N., and Sudeep Sarkar. "A perceptual prediction framework for self supervised event segmentation." Proceedings of the IEEE/CVF Conference on Computer Vision and Pattern Recognition. 2019.
>
> *We hope to have addressed all your questions, and look forward to any further discussions*

---

> > ### Comment · Reviewer_egXC · 2023-08-21
> >
> > Thank you for the rebuttal response with the clarifications requested -- I think that this is helpful and I'm currently still leaning towards maintaining my preliminary positive rating, pending final AC-reviewer discussion. A couple of notes/thoughts:
> > - Regarding the 1/3 layer protocols in (question 2), it would be helpful to indicate/clarify how other model designs may make fair comparisons with this work (since these seem specific to the proposed model design), so that future work can build constructively.
> > - For (question 3), if there is a baseline/ablation experiment that can show this difference not being significant, that can help to further strengthen the paper.

---

> > > ### Author Response · Authors · 2023-08-21
> > >
> > > Thank you for your discussion and the positive rating. Your comments have helped us refine the paper.
> > >
> > > * For any model generating a hierarchical structure of events, the proposed Hierarchical Level Reduction (HLR) algorithm could be applied for comparison and evaluation. The algorithm and pseudocode are provided in the supplementary material and we will release the code for it upon acceptance. On the other hand, methods that generate a single layer of event segments can directly compare to our 1-layer evaluation (as is common practice in all prior works) in Table 1. So far, existing segmentation models only output a single layer, HLR allows for more generalized evaluation of complex hierarchical structure of events for future works. We will clarify this in the camera ready submission.
> > > * Yes, we will evaluate the other methods with our pretrained CNN backbone on Ego4D+Epic and include the results in the camera ready submission. To get a rough idea, LSTM+AL in Table 1 has been trained on Ego4D+Epic, yet we still outperform it, showing the effectiveness of the hierarchical processing against single layer approaches.

---

### Official Review · Reviewer_L8go · 2023-07-08

**Soundness:** 3 good
**Presentation:** 2 fair
**Contribution:** 3 good
**Rating:** 4
**Confidence:** 3

**Summary:**

This paper introduces a self-supervised method to learn the hierarchical representation of streaming video events through predictive learning. This method segment videos into multiple chunks based on different semantic scale, from low-level fine-grained segmentation to high-level conceptual understanding. Meanwhile, the low-level boundaries should be consistent with the high-level ones. The authors apply predictive coding to model the causal structure of events, where the distance of the nearby video contents of variable contextual sizes decides the boundary score. In experiments, the model was trained on a random portion of the Ego4d and Epic-Kitchen datasets jointly, and it shows superior performance on the remaining video of the Epic-Kitchen dataset.

**Strengths:**

1. The author proposed a novel mode architecture. Unlike the widely-used existing deep learning architectures, it encodes the event hierarchy in the model, while the core, prediction block, aims at both event demarcation and cross-layer communication. Therefore, the model allows the representation to utilize information from higher and lower layers to predict the next input for accurate predictive coding.

2. Experiments are conducted on the union of two large-scale and challenging egocentric datasets, Ego4D and Epic-Kitchen, and the method is compared with sufficient state-of-the-art.

3. The model is more lightweight than its competitors. As Table 1 shows, a 4-layer CNN autoencoder using the proposed models is already better than the state-of-the-art.

**Weaknesses:**

1. The paper works on a better model for understanding video streaming. However, it only conducted experiments on egocentric video datasets. It is suggested to validate the algorithm on long from exo-centric video datasets, such as MovieNet [A] and the most recent NewsNet [B].

2. The proposed Hierarchical Gradient Normalization is a key component of the model optimization process. In the ablation experiment, we can see adjusting the reach of influence by gradient scaling improves the segmentation performance, but more experimental results are expected. The paper will have higher soundness when compared to different methods in this normalization process.

3. The paper mentions that the federated learning method is applied during the optimization process, but the implementation is more like simple distributed learning. In federated learning, the local datasets are usually in a different data domain/distribution, which is not addressed in the paper. In addition, is it possible to enable batch-wise training on a single machine, like a modern deep-learning model usually does?


[A] Huang, Qingqiu, Yu Xiong, Anyi Rao, Jiaze Wang, and Dahua Lin. "Movienet: A holistic dataset for movie understanding." In Computer Vision–ECCV 2020: 16th European Conference, Glasgow, UK, August 23–28, 2020, Proceedings, Part IV 16, pp. 709-727. Springer International Publishing, 2020.
[B] Wu, Haoqian, Keyu Chen, Haozhe Liu, Mingchen Zhuge, Bing Li, Ruizhi Qiao, Xiujun Shu et al. "NewsNet: A Novel Dataset for Hierarchical Temporal Segmentation." In Proceedings of the IEEE/CVF Conference on Computer Vision and Pattern Recognition, pp. 10669-10680. 2023.

**Questions:**

See weakness.

**Limitations:**

The paper already discussed its limitations.

---

> ### Author Rebuttal · Authors · 2023-08-08
>
>
> ### Weaknesses:
>
> 1. Thank you for the suggestion. Although NewsNet and MovieNet are large datasets with long enough videos, they have not yet been released. NewsNet was published very recently at CVPR23; it is not clear when it will be released to the public. Furthermore, in addition to the unreleased MovieNet dataset, it appears that the annotations do not contain action segments, but only coarse scene segments. We have included a table in the global response summarizing the statistics of relevant datasets.
> 2. Assuming that you are suggesting to apply our hierarchical normalization process to other methods for comparison, our normalization process was designed particularly for learnable hierarchical model, which we are the first to build. Currently there are no such models on which we can apply this method for comparison. Other methods like TW-FINCH [1] use hierarchical clustering but do not have any gradient updates for normalization. Learnable methods like LSTM+AL [2] only build a single layer of event predictions, therefore they do not need hierarchical gradient normalization. That being said, our proposed normalization scheme can be applied to any future work that is designed to model hierarchical events in a streaming manner.
> 3. Thank you for the remark, we will change the terminology to distributed learning. While it is possible to run on a single machine, it is not possible to enable batch-wise training due to the online event demarcation process and the streaming processing of videos. Different layers in the hierarchy receive updates at different times depending on the video contents and predictive capabilities of the model, therefore the parameters cannot be updated at the same time for all models/streams.
>
> [1] Sarfraz, Saquib, et al. "Temporally-weighted hierarchical clustering for unsupervised action segmentation." Proceedings of the IEEE/CVF Conference on Computer Vision and Pattern Recognition. 2021.
>
> [2] Aakur, Sathyanarayanan N., and Sudeep Sarkar. "A perceptual prediction framework for self supervised event segmentation." Proceedings of the IEEE/CVF Conference on Computer Vision and Pattern Recognition. 2019.
>
> *We hope to have addressed all your questions, and look forward to any further discussions*

---

> > ### Comment · Reviewer_L8go · 2023-08-18
> >
> > Dear authors, thanks for your response to the review and comparison of available long form video datasets.
> >
> > According to the table, although exo-centric videos have a wide range of applications, the proposed STREAMER method cannot be applied to those videos. Thus, it is hard to estimate the model performance on the important video domain. In another word, we cannot thoroughly validate the quality of the video event representations.

---

> > > ### Author Response · Authors · 2023-08-18
> > >
> > > Dear reviewer L8go, thank you for your reply.
> > >
> > > > According to the table, although exocentric videos have a wide range of applications, the proposed STREAMER method cannot be applied to those videos.
> > >
> > > 1. We would like to clarify that this method can be applied to exocentric video domain. However, as noted in the limitation, this model requires large datasets to better model long-term events. As of now, only egocentric domain has long enough videos to satisfy this requirement.
> > >
> > > > Thus, it is hard to estimate the model performance on the important video domain.
> > >
> > > 2. We understand your point about exocentric being an “important video domain”. However, egocentric is considered equally important and more suitable for vision learning in many applications, such as robotics, autonomous driving, etc. Furthermore, egocentric videos naturally incorporate human intentions, actions, and reactions, thus improving the model's ability to understand and predict human behavior [9 p1]
> > >
> > > 3. Egocentric learning faces more challenges than exocentric learning [1 p1; 4 p1] due to rapid viewpoint motions, varying lengths and lighting conditions, poor recording quality in egocentric videos, etc. [1 p1; 4 p1]. Therefore it is emphasized that **“egocentric action recognition [is] a research field on its own, apart from [exocentric] vision research”**,[4 p1] and works in the literature experimenting and **validating only on a single egocentric dataset** (i.e., EPIC-KITCHENS) **are generally accepted by the community** [5,6,7,8]. Unsupervised clustering-based approaches have already proven effective on the simpler setting of exocentric videos by utilizing precomputed features; we focus on the more challenging egocentric videos and show that we outperform the SOTA approaches.
> > >
> > > 4. In addition, humans learn from the egocentric viewpoint, and robots and other similar vision-powered agents should learn in a similar manner,[3 p.635; 2] because egocentric videos contain more information [3 p.634] and is beneficial to learn about the environment from the subjects' viewpoints.[2]
> > >
> > > > In another word, we cannot thoroughly validate the quality of the video event representations.
> > >
> > > 5. We have validated the quality of the video event representations (Tab. 2b; Fig. 4b; Supp. Figs. 3, 4) by evaluating them on the downstream task of event retrieval, as well as demonstrating SOTA hierarchical segmentation performance which solely relies on rich underlying representations (Tab. 1; Figs. 1, 4a; Supp. Figs. 1, 2; Supp. Video).
> > >
> > > References:
> > >
> > > [1] Wang, Xuanhan, Lianli Gao, Jingkuan Song, Xiantong Zhen, Nicu Sebe, and Heng Tao Shen. "Deep appearance and motion learning for egocentric activity recognition." Neurocomputing 275 (2018): 438-447.
> > >
> > > [2] Kanade, Takeo, and Martial Hebert. "First-person vision." Proceedings of the IEEE 100.8 (2012): 2442-2453.
> > >
> > > [3] Kang, Soo-Han, and Ji-Hyeong Han. "Video captioning based on both egocentric and exocentric views of robot vision for human-robot interaction." International Journal of Social Robotics (2021): 1-11.
> > >
> > > [4] Núñez-Marcos, Adrián, Gorka Azkune, and Ignacio Arganda-Carreras. "Egocentric vision-based action recognition: A survey." Neurocomputing 472 (2022): 175-197.
> > >
> > > [5] Cartas, Alejandro, et al. "Seeing and hearing egocentric actions: How much can we learn?." Proceedings of the IEEE/CVF international conference on computer vision workshops. 2019.
> > >
> > > [6] Wu, Yu, et al. "Learning to anticipate egocentric actions by imagination." IEEE Transactions on Image Processing 30 (2020): 1143-1152.
> > >
> > > [7] Tushar Nagarajan, Kristen Grauman: “Shaping embodied agent behavior with activity-context priors from egocentric video”, NeurIPS 2021.
> > >
> > > [8] Price, Will, Carl Vondrick, and Dima Damen. "Unweavenet: Unweaving activity stories." Proceedings of the IEEE/CVF conference on computer vision and pattern recognition. 2022.
> > >
> > > [9] Rodin, Ivan, et al. "Predicting the future from first person (egocentric) vision: A survey." Computer Vision and Image Understanding 211 (2021): 103252.

---

### Official Review · Reviewer_NKYW · 2023-07-09

**Soundness:** 3 good
**Presentation:** 3 good
**Contribution:** 3 good
**Rating:** 6
**Confidence:** 4

**Summary:**

This paper presents an algorithm for unsupervised temporal segmentation of videos in an online setting, by modeling actions hierarchically. The paper is based on the idea that prediction errors arise when a new action comes up, which makes the existing context insufficient (within a certain abstraction level). When a prediction error occurs, a new hierarchical level is open that encompasses the previous event as well as the new event that is just starting.

**Strengths:**

1. The idea in the paper is intuitive and clean. Both in terms of hierarchically representing the video, but also as a way of learning that hierarchy. Additionally, it is a good way of modeling long-term video (in terms of the potential memory problems that may arise), by encoding the past in a hierarchical way.

2. The quantitative results are better than baselines.

3. Good visualization videos in the supplementary. This kind of unsupervised tasks are hard to evaluate, both qualitatively and quantitatively. While the qualitative videos shown are probably cherrypicked and some times it is hard to figure out what the predicted divisions may mean, they make more sense than I would expect in such an unsupervised setting.

**Weaknesses:**

1. The method is not very clear from the figures or the explanation. I had to assume several things before I understood the idea, which I believe is simple and intuitive (which is good) enough to be explained in simpler terms. Because of that, I am not sure my understanding of the paper is completely correct. In the figures, where is the input video? There are two full figures showing the method and it is still not clear.

2. About the idea of “changing event” when there is a prediction error: this assumption is not idea. If the prediction model is trained to be as good as possible, wouldn't it be reasonable to predict a subevent that corresponds to the next event? Is the paper assuming that that kind of prediction is impossible?

3. The way the method makes predictions is (if I understand it correctly) cheating. They evaluate each level separately, and then select the level that best fits the ground truth (but they do so using ground truth). Ideally, a level should be selected *before* looking at the ground truth.

4. Evaluation: the qualitative videos are good, but it would be ideal to have better explainability results or experiments. It is not clear that the method is actually outputting semantically reasonable hierarchies. For example, having human studies evaluating the hierarchies, or evaluating on hierarchical datasets (like Finegym).

**Questions:**

1. There are only 3 layers in the results. Does this mean the algorithm stops opening layers when it reaches the third level?

2. At the beginning of training the first layers will be bad (untrained), so the “events” will be shorter (prediction errors will come sooner). As training evolves, layer X will have to model higher and higher level events. This can be a waste of training at the beginning, because the layers will end up modeling abstraction levels that they were not modeling initially. Have the authors considered to start training just the first layer; when it stabilizes, start training second layer (along with first layer); etc.?

3. Weakness 2 is formulated as a question, so I am referencing it here.

4. Are the neural networks for every layer different? Is a new neural network created for every layer?

5. It would be good to have as a reference some supervised method. How far is this from a model with more direct supervision? This is in line with the general question (that would be great to get the authors' opinion on) of "is self-supervision the answer to modeling hierarchies in videos?" (as opposed to using some sort of supervision).

Overall, the paper presents an intuitive and interesting idea. However, there are important weaknesses that need to be addressed. I believe the paper in the current form is not ready to be published, but that it can be with a couple of iterations during the rebuttal period. Therefore, expecting these weaknesses to be addressed, I recommend acceptance of the paper.


**Limitations:**

Limitations and broader impact are properly addressed.

---

> ### Author Rebuttal · Authors · 2023-08-08
>
> Thank you for your general remarks on the idea and pointing out its strengths, such as avoiding memory constraints in long-term events modeling.
>
> ### Strengths:
>
> 3. Thanks for this remark. We just wanted to clarify that the qualitative videos were not cherrypicked, but selected at random.
>
> ### Weaknesses:
>
> 1. Thank you for the suggestion, we have modified the figures to be simpler and we will include them in the camera ready version. We have included the modified figures in the pdf of the global response.
> 2. Yes, we assume that perfect high-detailed long-term prediction is not possible for any of the layers, because the higher levels in the hierarchy allow for longer prediction of events by sacrificing finer details, whereas lower levels can predict high-detailed short-term events. If a well-trained layer is capable of predicting the future event given its event model, then the predicted event indeed belongs to the current event. This predicted event is however represented as a subevent in the layer below it.
> 3. Our architecture outputs a rich representation of events showing a hierarchical structure; however, the groundtruth annotations only provide a single layer of events. This should be viewed as a shortcoming of the existing annotations, rather than a weakness in the output of our model. Therefore, all other unsupervised approaches have used the groundtruth in their evaluation to match their models' output to a specific segmentation granularity provided by the groundtruth. For instance, ABD [1] iteratively performs refinement on the outputs to merge neighboring events until the number of output events matches the groundtruth events, TW-FINCH [2] has a similar procedure as a clustering stopping criterion. Unlike other methods, STREAMER outputs a rich hierarchical representation, then we perform evaluation with best layer (or greedily using our proposed hierarchical layer reduction algorithm).
> 4. We agree that the event segmentation quantitative results only validate the quality of boundary localization and does not explicitly validate the semantic quality of the hierarchy. As you have mentioned in the strengths (point 3), it is challenging to evaluate these kinds of unsupervised tasks; there are no agreed upon evaluation procedures/metrics for assessing the quality of a hierarchy. However, we want to point out that STREAMER restricts the output hierarchy to have aligned boundaries, the higher layers only filter out some boundaries in the lower layers based on the prediction errors. Therefore, we can deduce that if boundaries at all layers seem reasonable, the hierarchy have valid semantic relationships between layers. This kind of constraint is better than a supervised approach attempting to predict three different layers independently, as their boundaries would not align. The videos in FineGym are, on average, 50 seconds; it is therefore not possible to learn high level events with long-term temporal dependencies from such short videos. We have included a table in the global response summarizing the statistics of relevant datasets, all exhibiting the same limitation.
>
> ### Questions:
> 1. No, STREAMER automatically adds as many layers as needed, and this is determined by the amount of data available for training. For illustration and experiments, we show three layers, but we are not limited by three.
> 2. Yes, you are correct. This is in fact how we are training STREAMER: we start by training one layer, after it has trained and stabilized, another layer is created and trained simultaneously. This is described in L28:30, L144:146, and L223:224.
> 3. Please see above.
> 4. Yes, a completely new layer is initialized; however, all layers have the same architecture and loss function.
> 5. To the best of our knowledge, there are no supervised methods for hierarchical event segmentation on EPIC-KITCHENS datasets. In answer to the second half of the question, yes, we believe self-supervision is the solution to hierarchical modeling of events, for the following reasons:
>     - Creating hierarchical annotations manually is much more expensive than a single layer of annotations. This is clearer when considering other 2D perceptual signals, such as images. Creating multiple levels of object mask annotations (or even bounding boxes) for objects is not scalable.
>     - Hierarchical annotations can be subjective leading to inconsistent events. For example, adding salt and pepper, in a cooking video, can be represented by one person as two different high level events (with low level being grasp, shake, etc.), while a different person may consider adding salt and pepper a single event at the lowest level. Additionally, annotators may disagree on the relative level assignment between events (e.g., is "adding salt" at the same level as "chopping vegetables"?). Multiple different and semantically correct hierarchies may co-exist.
>     - We believe co-occurring patterns should be discovered using self-supervision such that discovered patterns can be used as building blocks for other higher-level patterns in a compositional structure. Forcing supervised learning on this structure can lead to poorer features that satisfy the given annotations without considering how intermediate features can be used for other higher-level tasks.
>     - It is not practical to consider all different events and patterns at all levels of the hierarchy during the manual creation of hierarchical annotations. A self-supervised approach such as STREAMER can detect all different patterns and learn semantic representations for each of them.
>
> [1] Du, Zexing, et al. "Fast and unsupervised action boundary detection for action segmentation." CVPR2022.
>
> [2] Sarfraz, Saquib, et al. "Temporally-weighted hierarchical clustering for unsupervised action segmentation." CVPR2021.
>
> *We hope to have addressed all your questions, and look forward to any further discussions*

---

### Author Rebuttal · Authors · 2023-08-08

We thank the reviewers for their comments and useful suggestions. We think that these comments have helped us better refine the paper.

A common concern expressed in the reviews is evaluation only on egocentric dataset. We address this concern here in addition to individual rebuttals.

Our method, STREAMER, is a self-supervised architecture that relies on predictive learning for hierarchical segmentation. In our model, higher-order predictive layers receive sparser learning signals than lower-order layers (see Figure 2 in the attached pdf), because the first layer directly predicts frames, whereas higher layers only receive events that cannot be predicted at lower levels. Short videos do not allow for higher order predictions and learning long-term temporal dependencies. Therefore, training higher levels in the hierarchy requires a large dataset (i.e., total number of hours) and longer videos (i.e., average video duration) in order to model high-level events. These requirements constraint the choice of datasets on which we can run and evaluate STREAMER.


| Dataset               | Egocentric    | Total Hours | Average Duration (minutes) | Large Datasets | Long Videos | Comments                                           |
|-----------------------|---------------|-------------|----------------------------|----------------|-------------|----------------------------------------------------|
| Ego4D                 | ✅            | 3670        | 23                         | ✅             | ✅          | Large dataset and long videos                      |
| EPIC-KITCHENS 55      | ✅            | 55          | 10.5                       | ✅             | ✅          | Large dataset and long videos                      |
| EPIC-KITCHENS 100     | ✅            | 100         | 8.5                        | ✅             | ✅          | Large dataset and long videos                      |
| GTEA                  | ✅            | 0.58        | 1.23                       | ❌             | ❌          | Small dataset and short videos                     |
| Breakfast Action      | ❌            | 77          | 2.3                        | ✅             | ❌          | Short videos                                       |
| YouTube Instructional | ❌            | 5           | 2                          | ❌             | ❌          | Small dataset and short videos                     |
| Hollywood Extended    | ❌            | 3.7         | 0.23                       | ❌             | ❌          | Small dataset and short videos                     |
| 50 Salads             | ❌            | 4.5         | 4.8                        | ❌             | ❌          | Small dataset and short videos                     |
| FineGym               | ❌            | 708         | 0.91                       | ✅             | ❌          | short videos                                       |
| ActivityNet Captions  | ❌            | 849         | 2.5                        | ✅             | ❌          | short videos                                       |
| MovieNet              | ❌            | 2174        | 117                        | ✅             | ✅          | Unreleased videos and contains only scene segments |
| NewsNet               | ❌            | 946         | 57                         | ✅             | ✅          | Unreleased dataset                                 |




Based on our review of available datasets for both ego-centric and exo-centric settings, as shown in the table above, we found that many of the available datasets (typically used in event segmentation) do not provide long enough videos. MovieNet and NewsNet are both large datasets with long videos but have not yet been released to the public. In addition, MovieNet does not contain action segments; it contains coarse scene segments. Whereas, the NewsNet paper was published very recently in CVPR23. The only available options to train and evaluate STREAMER is large-scale ego-centric datasets, where the available datasets provide large enough scale (i.e., total number of hours) with a long average duration per video for streaming and high-order temporal prediction. We plan to add this discussion to the supplementary paper.

---

### Decision · Program_Chairs · 2023-09-21

**Decision:**

Accept (poster)

**Comment:**

This paper received mixed ratings: 2 borderline rejects, 1 borderline accept, and 1 weak accept. The reviewers concurred that the paper presents an intriguing and innovative idea. In particular, they found the concept of learning to represent videos in a hierarchical fashion to be intuitive, and they regarded the proposed architecture as novel. Additionally, they appreciated the generally well-written nature of the paper.

The most significant criticism, primarily from `L8go` and `s9dg`, was that the experiments were exclusively conducted on egocentric video datasets (EPIC-KITCHENS and Ego4D) and not on other "exo-centric" datasets that are perhaps more widely used in the video understanding community. Because of the limited evaluation scenario, the reviewers raised questions about the generality of the proposed approach.

In their rebuttal, the authors argued that there are no other publicly available datasets of comparable scale and duration. They provided a table summarizing the current video datasets, emphasizing that EPIC-KITCHENS and Ego4D are only two publicly available datasets that provide both the scale and long duration.

After the rebuttal, the two reviewers `NKYW` and `egXC` concurred with the rebuttal, acknowledging that there were limited alternatives for the authors to utilize. The other two reviewers, `L8go` and `s9dg`, remained skeptical, re-iterating their core concern that the proposed approach has not been evaluated on exo-centric videos that have a wide range of applications.

During the post-rebuttal discussion phase, reviewers `NKYW` and `egXC` reiterated their recommendation to accept the paper, arguing that the paper's strengths outweighed the weakness of focusing on EPIC-KITCHENS and Ego4D for experiments. Reviewers `L8go` and `s9dg` did not participate in the discussion. This meta-reviewer found the arguments by `NKYW` and `egXC` to be persuasive. They are also well-aligned with the four reviewers' consensus that the paper presents an intriguing idea with a novel architecture. In addition, this meta-reviewer agrees with the authors that 1) the limited availability of existing exo-centric datasets (as was clearly shown in the dataset table provided in the rebuttal) makes it infeasible to evaluate on other datasets as the reviewers `L8go` and `s9dg` requested, and 2) that egocentric video understanding is equally important as exo-centric video tasks and has numerous real-world applications (which was argued in response to `L8go`).

After careful consideration, we are happy to recommend acceptance. In the final version, we strongly recommend that the authors provide thorough explanations regarding the available large-scale, long-form video datasets (or lack thereof).